# Effects of Sucrose Feeding on the Quality of Royal Jelly Produced by Honeybee *Apis mellifera* L.

**DOI:** 10.3390/insects14090742

**Published:** 2023-09-04

**Authors:** Ying Wang, Lanting Ma, Hongfang Wang, Zhenguo Liu, Xuepeng Chi, Baohua Xu

**Affiliations:** College of Animal Science and Technology, Shandong Agricultural University, Tai’an 271018, China; wangying@sdau.edu.cn (Y.W.); malanting@sdau.edu.cn (L.M.); whf@sdau.edu.cn (H.W.); zgliu@sdau.edu.cn (Z.L.); xixuepeng@sdau.edu.cn (X.C.)

**Keywords:** sucrose feeding, *Apis mellifera* L., nutritional compositions, royal jelly, stored food

## Abstract

**Simple Summary:**

Royal jelly (RJ) is a nutritious substance secreted by the hypopharyngeal glands (HPGs) of bees, and its nutritional composition may be influenced by feeding factors. There is controversy regarding the relationship between sucrose feeding and the quality of royal jelly. Comparisons were made between RJ obtained from sucrose-fed colonies and honey-fed groups. The results showed that sucrose-fed colonies had higher levels of certain amino acids in the RJ, but no significant differences were found in terms of moisture, protein, glucose, minerals, or other amino acids. Sucrose feeding did not affect the activity of sucrase or the development of the HPGs in nurse bees. Stored food samples from sucrose-fed colonies had higher sucrose levels compared to sealed combs and natural honey. Natural honey had different levels of moisture, Ca, Zn, and Cu compared to honey from sucrose-fed colonies. Overall, sucrose feeding had a minimal impact on the major components of RJ. This study provides important parameter information for further understanding the impact of sucrose feeding on the quality of RJ.

**Abstract:**

Royal jelly (RJ) is a highly nutritious secretion of the honeybees’ hypopharyngeal glands (HPGs). During RJ production, colonies are occasionally subjected to manual interventions, such as sucrose feeding for energy supplementation. This study aimed to assess the impact of sucrose feeding on the composition of RJ. The results indicated that RJ obtained from sucrose-fed colonies exhibited significantly higher levels of fructose, alanine, glycine, tyrosine, valine, and isoleucine compared to the honey-fed group. However, no significant differences were observed in terms of moisture content, crude protein, 10-HDA, glucose, sucrose, minerals, or other amino acids within the RJ samples. Moreover, sucrose feeding did not have a significant effect on midgut sucrase activity, HPGs development, or the expression levels of *MRJP1* and *MRJP3* in nurse bees. Unsealed stored food samples from sucrose-fed bee colonies demonstrated significantly higher sucrose levels compared to sealed combs and natural honey. Additionally, natural honey exhibited higher moisture and Ca levels, as well as lower levels of Zn and Cu, in comparison to honey collected from bee colonies fed sucrose solutions. Based on these findings, we conclude that sucrose feeding has only a minor impact on the major components of RJ.

## 1. Introduction

Royal jelly (RJ) has long been recognized as a valuable functional food owing to its wide range of potential health benefits. This yellowish, viscous secretion is produced by the hypopharyngeal and mandibular glands of young worker bees and has been used in commercial medical products, health foods, and cosmetics [1,2]. The composition of RJ is complex, comprising water, proteins, sugars, fat, ash, vitamins, and a significant number of bioactive substances [2,3]. Its functional properties have been shown to include antibacterial, anti-inflammatory, antioxidant, anti-cholesterol, and anti-tumor effects [2].

China is the largest producer and exporter of RJ, with its output accounting for over 90% of the world’s RJ output [4]. In China, beekeepers usually produce RJ via two production methods: migratory beekeeping production and beekeeping in permanent apiary production. As most of China has a temperate climate with four distinct seasons, most beekeepers use migratory beekeeping to chase flowers for nectar to obtain high RJ yields and minimize breeding costs, tracking the successive flowering of Chinese pollen source plants from south to north. However, high labor and transport requirements, and the fact that many beekeepers continue to keep bees on fixed sites, mean that during periods when nectar is scarce outside, feeds such as honey or cane sugar must be provided to meet the carbohydrate needs of the colony.

Research has shown that various dietary types, bee races, botanical origin, and beekeeping management can affect the quality and composition of RJ, including proportions of protein, fat, and 10-hydroxy-trans-2-decenoic acid (10-HDA) [5,6,7]. Recent studies have demonstrated that different floral resources during migratory beekeeping can significantly influence the chemical composition and antioxidant activity of RJ [8]. Previous studies have shown that the results of artificial feeding, especially sucrose feeding, on RJ production and quality is unstable [9,10,11,12]. Furthermore, most of these studies have focused on the effects of artificial feeding on the fructose, glucose, and sucrose content of royal jelly and little is known about the global effects of artificial feeding on royal jelly quality. Another major concern about artificial feeding during RJ production is potential quality and safety issues. While it is unclear whether queen larvae are directly fed with honey by nurse bees, it has been confirmed that nurse bees may feed them with beebread [13], meaning that substances in food, such as heavy metals and pesticides, may directly affect RJ quality. However, currently few studies have researched the effects of feed on trace element content in royal jelly.

Given the potential value of RJ as a functional food, it is critical to establish detailed knowledge of its safety control and quality management. As the world’s largest producer of RJ, China’s beekeeping practices and feeding methods may have significant implications for the quality and composition of RJ. In this study, we systematically evaluated the effects of sucrose feeding on the chemical composition and quality of RJ using conventional chemical analysis methods combined with metabolomics techniques. In addition, we attempted to reveal the mechanism of effect of sucrose feeding on RJ quality by examining the transformation of sucrose in the honeycomb and the expression characteristics of genes in the head of honeybees fed with sucrose. This study is a step towards understanding the effects of artificial feeding on RJ production and quality, providing insights into factors that may impact the safety and efficacy of this valuable functional food.

## 2. Materials and Methods

### 2.1. Bees

All experiments were conducted from August to October of 2021 at the Shandong Agricultural University (117.12° N, 36.20° E) in Tai’ an, Shandong Province, China. During this period, there were no external sources of nectar available; however, there was a season of blooming plants that provided pollen sources. Thus, all colonies had ample pollen supply throughout the entire experimental period.

Ten sister-queen colonies of *Apis mellifera* L. were used in this study. The colonies contained similar numbers of bees and broods (eight frames in the brood chamber and five frames in the super) and were randomly allocated to two groups (five colonies per group).

### 2.2. Feeding Treatment

To replicate a natural situation, at the beginning of the study, combs with stored honey were all removed using a honey extractor. The control group was provided with 1.5 kg pure honey (*Vitex negundo* var. *heterophylla*) daily, and the experimental group was fed with 1.5 kg sucrose syrup (50 wt. %) daily. The pre-feeding period for these two groups lasted 30 days. The study period was approximately 6 weeks.

### 2.3. RJ Production

The RJ was collected starting at the end of pre-feeding. During the experimental period, we continuously collected and counted the production of 10 batches of royal jelly. For each batch, 315 1-day-old larvae were transferred into the queen cells. All RJ samples were collected 72 h after grafting using the method described by Wang et al. [14], and stored at −40 °C until further testing.

### 2.4. Stored Food Collection

To determine whether sucrose feeding could directly affect the composition of stored food, feed samples were collected separately at the end of the experiment from both the sealed and unsealed combs of bee colonies fed sucrose (these feed samples are referred to as sealed stored food [SSSF] and unsealed stored food [SUSF], respectively). The stored food samples were collected and stored at −20 °C until further testing. The nutritional components of moisture, fructose, glucose, sucrose, elements of pure honey (PH), SSSF, and SUSF were analyzed. 

### 2.5. Analytical Procedures of RJ and Stored Food

To calculate the production performance of RJ when the RJ producing frame was taken out from the hive, the number of queen cells containing RJ were counted, and the acceptance rate of the queen cells were calculated. The RJ production of each RJ producing frame was weighed (accurate to 0.1 g) and recorded as the RJ production of each colony. During the experimental period, the RJ production of each colony was counted 5 times and the average value was calculated as the unit yield of RJ.

To determine the physicochemical characteristics of RJ and comb-feed, a 0.5 g RJ sample or 5 g comb feed sample (to the nearest 0.1 mg) was used for each test. The standard methodology of prior literature was used to evaluate the moisture, crude protein, 10-HDA [14], amino acids [15], and minerals [16]. The glucose, sucrose, and fructose were quantified by the HPLC method reported by Sevgi et al. [17] using an HPLC machine (Waters 515 equipped with a Waters 450 refractive index detector and a XBridge Amide column [4.6 mm × 250 mm, 3.5μm]).

### 2.6. Scanning Electron Microscopy Analysis

To compare the development of the HPGs, a total of 10 9-day-old workers were selected for each group. After dissecting the pharyngeal glands, a scanning electron microscopy (SEM) observation was conducted following the methodology of Jianke et al. [18]. Furthermore, the developmental levels of the HPGs were assessed and classified according to the method described by Wang et al. [19].

### 2.7. Determination of Midgut Sucrase Enzyme Activity

For each sample, the midguts of ten 9-day-old workers were dissected for sucrase activity measurement. The midguts were placed in a 1.5 ml polypropylene centrifuge tube and weighted. Then PBS buffer solution (0 °C) in a 1:9 (weight: volume) was added and the midguts were homogenated under ice bath conditions. This was centrifuged at 4500 r/min for 10 min and then the supernatant was removed. The determination of sucrase activity in the midgut was carried out using the Sucrase Assay Kit (Nanjing Jiancheng Bioengineering Institute, Nanjing, China) following the manufacturer’s instructions.

### 2.8. RNA Extraction, cDNA Synthesis, and Reverse Transcription Quantitative PCR

The total RNA was extracted from HPGs of 15 9-day-old workers using a RNAiso Plus kit (Takara Beijing, China) according to a standard protocol. cDNA was obtained from 1 μg total RNA by reverse transcription using a Transcript All-in-One First-Strand cDNA Synthesis SuperMix (TransGen Biotech, Beijing, China) following instructions. Reverse transcription quantitative PCR (qRT-PCR) was carried out on a ABI 7500 Real Time PCR System (Applied Biosystems, Waltham, MA, USA) with an SYBR PrimeScript RT-PCR kit (Takara, Beijing, China). The transcript levels of genes were quantified using the 2^−ΔΔCT^ method. The primers used for qRT-PCR are listed in Appendix A.

### 2.9. Statistical Analysis

For each treatment, five samples were used and all the assays were carried out in quintuplicate. Data analysis was performed using an SPSS statistical software package (version 21.0; SPSS Inc., Chicago, IL, USA). For the comparison between the two groups, an independent-samples *t*-test was used. Prior to data analysis, a normality test was conducted, and, after confirming homogeneity of variances, a One-Way ANOVA with Tukey’s HSD test was performed to compare the carbohydrate and mineral components among multiple groups. The data were reported as the mean ± SD, and significant differences were recognized at *p* < 0.05. 

## 3. Results

### 3.1. Unit Yield of Royal Jelly and the Acceptance Rate of the Queen Cells

To determine whether sucrose feeding could affect RJ production, we firstly examined the unit yield of RJ and the acceptance rate of the queen cells (Figure 1). There was no significant difference in the unit yield of RJ (329.0 ± 9.76 g for HF vs 321.2 ± 9.76 g for SF, one per colony; *p* = 0.351) and the acceptance rate of the queen cells (90.2 ± 3.3% for HF vs 89.4 ± 4.4% for SF, one per colony; *p* = 0.401) between the honey feed and the sucrose feed groups.

### 3.2. Conventional and Mineral Composition of the Royal Jelly

To explore the effect of sucrose feeding on the quality and nutritional value of RJ, we examined the main nutritional components of RJ samples (Figure 2). Compared to the honey-fed group, the sucrose-fed group exhibited significantly higher levels of fructose in RJ (*p* = 0.033). However, there were no significant differences observed in the concentrations of moisture, crude protein, 10-HDA, glucose, or sucrose in RJ samples between the two treatment groups (*p* = 0.157, 0.370, 0.962, 0.931, and 0.367, respectively).

The amino acid content of RJ samples are reported in Table 1. There was no significant difference in the content of the 12 amino acids in the royal jelly samples between the two treatment groups (all *p* > 0.05), except for five amino acids, namely Alanine, Glycine, Tyrosine, Valine, and Isoleucine, which were significantly higher in the sucrose-fed group than in the honey-fed group. 

The mineral content of the analyzed RJ samples is shown in Appendix A. No statistically significant differences were noted between the HF and the SF in the contents of Fe, Zn, Cu, Mn, Na, K, Ca, or Mg.

### 3.3. Conventional and Mineral Composition of the Stored Food

The proximate and mineral composition of the analyzed stored food samples are shown in Figure 3 and Appendix A, respectively. The moisture contents of the SUSF were significantly higher than honey and the SSSF (Figure 3B). Furthermore, the sucrose content of the stored food from unsealed combs of the sucrose fed groups were significantly higher than those collected from sealed combs of the experimental groups and control groups (Figure 3E). However, no statistically significant differences were observed in the contents of fructose and glucose among the three detected group samples (*p* = 0.292 and 0.054, respectively).

As shown in Appendix A, significantly higher amounts of Zn and Cu were found in the stored food samples in the sucrose fed groups (SSSF and SUSF) compared with honey (Appendix A, *p* = 0.005 and 0.002, respectively). Moreover, the stored food of honey fed groups had significantly higher levels of Ca than that collected from either sealed or unsealed combs in sucrose-fed colonies (Appendix A, *p* = 0.016). However, no significant differences were found in Fe, Mn, Na, K, or Mg contents among the three detected group samples (all *p* > 0.05).

### 3.4. Sucrose Enzyme Activity, Hypopharyngeal Gland Development and Gene Expression Analysis

To determine whether the sucrose enzyme activity is caused by different sugar diets, we examined the worker’s midgut sucrose enzyme activity on their 16th days. Bees of SF had significantly higher levels of sucrose enzyme activity than HF (Figure 4A, *p* < 0.01).

To investigate whether different sugar feeding has an effect on the development of the HPGs, we evaluated the developmental variations of HPGs between the HF and SF groups (Figure 4B). Scanning electronic microscopy (SEM) photographs showed that the HPGs of honeybees in both treatments were well developed (Figure 4B(a)). The statistical analysis of hypopharyngeal gland development grades showed that there was no significant difference in the development of royal jelly gland between the two groups (Figure 4B(b), *p* = 0.608).

Furthermore, we evaluated the effect of different diets on the gene expression of the *MRJP 1* and *MRJP 3* in the HPGs (Figure 4C,D). The qRT-PCR results showed that there was no significant difference in the expression of *MRJP 1* and *MRJP 3* in the HPGs of honeybees between the two groups (*p* = 0.578 and *p* = 0.896, respectively).

## 4. Discussion

The research and practices of animal nutrition have proved that feed quality does not only affect the quality of livestock products, but that it also directly affects the safety of livestock products [20]. In recent years, with the development of the feed industry, artificial bee feed products (such as pollen substitute, sucrose, fructose syrup, etc.) have been widely used and popularized in beekeeping. However, we still lack an adequate understanding of the impact of bee feed quality on the quality and safety of bee products.

Generally, fresh RJ comprises water (50–70%), proteins (9–18%), 10-HDA (>1.4%), fructose (3–13%), glucose (4–8%), and sucrose (0.5–2.0%) [2]. The results of the present study are consistent with those of previous studies. The component 10-HDA is the most important bioactive substance in royal jelly and sucrose is thought to play a key role in the biosynthesis of 10-HDA [21]. However, we found that the 10-HDA levels in RJ from sucrose-fed groups were not affected. This may have been because honey feeding and sucrose feeding both satisfied the carbon source demands required for 10-HDA synthesis. When external nectar and powder sources are abundant, honey bees will actively cultivate the queen cells. At these times, the nursing bees have the highest propensity to secrete royal jelly. Therefore, beekeepers often use this biological characteristic of bees to chase the flowering period to obtain a high yield of royal jelly. In this study, we found that in seasons with a lack of nectar, artificial sucrose feeding can also stimulate the propensity of nurse bees to secrete royal jelly in order to improve the acceptance rate of queen cells and, thus, promotes a high yield of royal jelly for beekeepers.

There are conflicting findings regarding the effects of carbohydrate feeding on RJ quality. Recent research has indicated that the carbohydrate composition of RJ, including glucose, fructose, sucrose, erlose, and raffinose, can be influenced by floral sources [8,22,23]. Different types of pollen feeding can impact the 10-HDA content [24]. Harvest time has been found to significantly influence the yield and chemical composition of RJ [10]. On the contrary, some studies have demonstrated that sugar cane feeding increases the sucrose and erlose content of RJ [10], while others have reported no influence on physicochemical parameters when using artificial sugar feeding [12]. In our study, sucrose feeding significantly increased fructose content but had no effect on glucose and sucrose levels. Furthermore, it is worth noting that other studies have also shown no impact of artificial supplementary feeding on the physicochemical and microbiological composition of RJ produced by *Africanized* bees [25]. Although RJ is a mixture secreted by the worker bee’s mandibular and HPGs, its quality is influenced by various factors making it unstable. Therefore, further research on the effects of feed components on RJ quality and safety, as well as investigations into the secretion mechanisms of RJ, will undoubtedly contribute to the standardized production of royal jelly.

Regardless, we should not underestimate the impact of feed ingredients on the safety of royal jelly since there is direct evidence that high levels of dietary nutrient elements can be concentrated in honey which may affect the quality of royal jelly in the food chain. An earlier study showed that the Zn content of royal jelly was positively affected by high levels of dietary Zn [26]. Xuepeng et al. [27] also found that supplementation with sodium selenium in sucrose solution could substantially improve royal jelly’s Se content. In this study, although sucrose feeding did not affect the mineral element contents of RJ, significantly higher amounts of Zn, Cu, and Ca were found in the stored comb feed samples of sucrose-fed groups. These excess minerals obviously came from the provided sucrose solution, but their content had not yet reached a threshold that would cause significant changes in the content of heavy metals in the royal jelly. However, since we did not analyze the mineral elements of water and sucrose, we could not ascertain the actual reason for this effect. A recent study showed that the content of mineral elements in RJ was significantly influenced by post-grafting time [28]. Moreover, compared with previous studies, the mineral contents of royal jelly produced in different countries or regions were quite unstable [28,29]. Further studies on the enrichment mechanism of minerals in royal jelly will be of great significance for a better understanding of the underlying causes of this difference and to better control the quality and safety of royal jelly production.

Previous studies have demonstrated a significant correlation between the activity of digestive enzymes and the digestive capacity of bees [14,30]. Sucrase, an important digestive enzyme responsible for the hydrolysis of sucrose in the midgut of honey bees, has received limited attention regarding its response to dietary nutrition in these insects. Our study reveals that the activity of midgut sucrase in honey bees was not influenced by varying levels of dietary fatty acids [31]. In the current investigation, we observed a pronounced increase in midgut sucrase activity in the group that was fed sucrose when compared to the group that received a honey-based diet. This finding implies that the expression and activity of sucrase are induced by the presence of sucrose within the dietary composition. Nevertheless, it remains to be determined whether the disparities observed in the activities of midgut digestive enzymes could account for the variations in fructose and specific amino acid levels within royal jelly between the two experimental groups. Consequently, further comprehensive investigations are warranted to explain this matter and allow a deeper understanding of the underlying mechanisms involved.

MRJP1 and MRJP3 are the two most abundant MRJPs in royal jelly [32]. Previous studies have demonstrated that the expression of the major royal jelly protein genes in *Apis mellifera* differs according to caste types, developmental stages, and tissues [33,34]. Among them, both *MRJP1* and *MRJP3* exhibit high levels of expression in the HPGs of nursing bees. Recent investigations have revealed that various feeding activities, such as bacterial ingestion, can considerably affect the expression levels of the *MRJPs* expression in the HPGs of worker bees [35]. Nevertheless, in the current study, no significant alterations were observed in the expression of *MRJP1* or *MRJP3*. This suggests that sucrose feeding may not induce the expression of these two genes.

Another concern is the possibility of royal jelly adulteration by inexpensive sweeteners which are used as substitutes for honey [16,36,37]. For queen larvae, although it is not clear whether nurse bees directly feed them with nectar or feed, it is known that nurse bees feed them a small amount of beebread in some cases [13]. Therefore, if the feed contains some harmful substances, there is a safety risk of polluting the queen bee quality. In this study, we found that the nurse bees may not directly feed the queen larvae with artificial feed because sucrose feeding did not lead to significant changes in the sucrose content of the royal jelly. However, another possibility may be that the nurse bees directly feed the queen larvae with transformed sugar food since sucrose is rapidly hydrolyzed to fructose and glucose under the action of the bee’s efficient invertase system. In any case, it is worth performing further research and exploration.

Additionally, in our study, all royal jelly samples were obtained from sister-queen colonies. This may explain why the compositions of the royal jelly obtained from the control and experimental groups were similar, while larger fluctuations were reported in previous studies [14,38]. Further investigation into the manufacturing factors that may affect royal jelly quality, especially the different honeybee varieties, feeding sources, seasonal factors, and regional origins, is very important.

## Figures and Tables

**Figure 1 insects-14-00742-f001:**
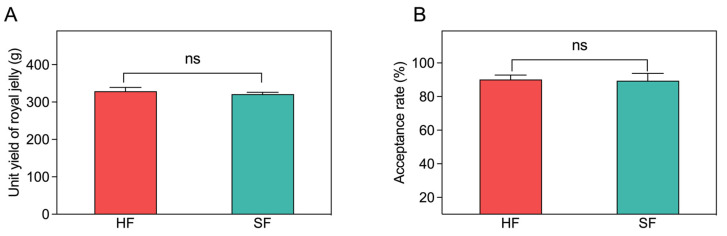
Effects of honey feeding and sucrose feeding on production of royal jelly (**A**) and the acceptance rate of queen cells (**B**). HF, honey feeding group; SF: sucrose feeding group. Values are means ± SD, *n* = 5. The independent samples *t*-test was adopted for comparisons between groups. ns, no significance (*p* > 0.05).

**Figure 2 insects-14-00742-f002:**
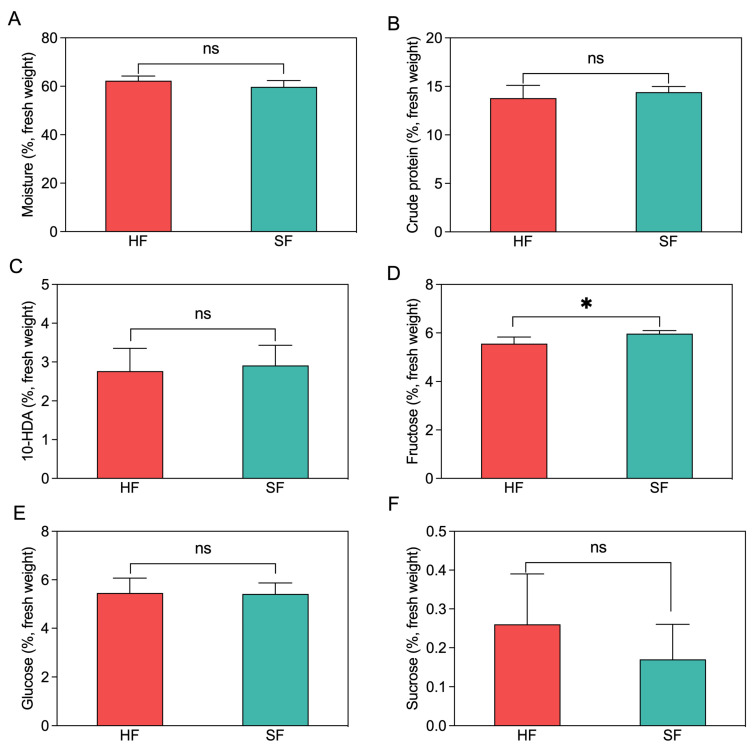
Proximate composition of the analyzed RJ samples obtained from honey feeding and sucrose feeding procedures (g/100 g, fresh weight). (**A**) The moisture content of RJ samples. (**B**) The crude protein content of RJ samples. (**C**) The 10-HDA content of RJ samples. (**D**) The fructose content of RJ samples. (**E**) The glucose content of RJ samples. (**F**) The sucrose content of RJ samples. HF, honey feeding group; SF, sucrose feeding group. Values are means ± SD. * *p* < 0.01 via independent samples *t*-tests, and ns, no significance (*p* > 0.05).

**Figure 3 insects-14-00742-f003:**
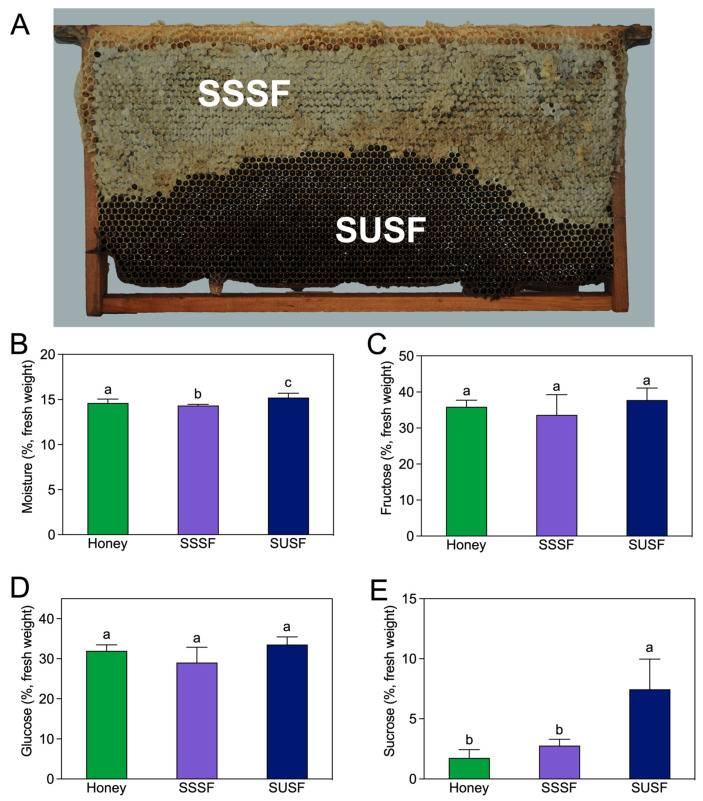
Comparison of proximate composition of stored food obtained from honey-fed and sucrose-fed colonies (g/100 g, fresh weight). (**A**) Sampling area diagram. (**B**) The moisture content of stored food samples. (**C**) The fructose content of stored food samples. (**D**) The glucose content of stored food samples. (**E**) The sucrose content of stored food samples. Honey, stored food obtained from honey-fed colonies; SSSF, stored food obtained from sealed combs of sucrose-fed colonies; and SUSF, stored food obtained from unsealed combs of sucrose-fed colonies. Values are means ± SD. Different lowercase letters indicate significant differences among treatments at the 0.05 level, according to Tukey’s HSD test.

**Figure 4 insects-14-00742-f004:**
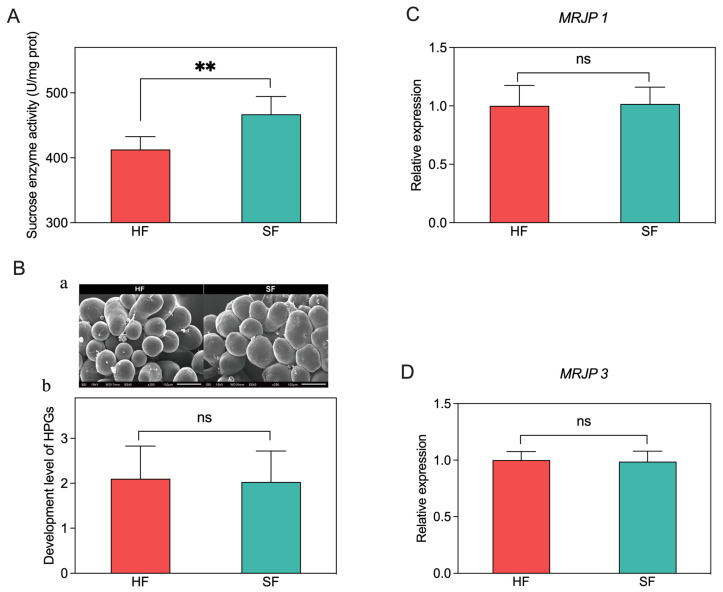
The effects of different sugar diets on the activity of sucrose enzymes (**A**) the development of HPGs (**B**) and the expression of the *MRJP 1* (**C**) and *MRJP 3* (**D**). (**a**), Morphology of HPGs (SEM, ×250); (**b**), comparison of the development levels of HPGs. HF, honey feeding group; and SF, sucrose feeding group. Values are means ± SD. ** *p* < 0.01 by independent samples *t*-test, and ns, no significance (*p* > 0.05).

**Table 1 insects-14-00742-t001:** Comparison of the amino acid (AA) composition (g/100 g, fresh weight) of RJ between honey-fed and sucrose-fed groups. HF, honey feeding group; and SF, sucrose feeding group. Values are means ± SD. * *p* < 0.05 using independent samples *t*-tests.

Amino Acid	Group		F	P
	HF	SF		
Phenylalanine	0.508 ± 0.029	0.534 ± 0.057	1.497	0.256
Alanine	0.328 ± 0.022	0.410 ± 0.062 *	10.923	0.011
Methionine	0.052 ± 0.019	0.117 ± 0.056	3.751	0.089
Glycine	0.350 ± 0.019	0.380 ± 0.053 *	8.533	0.019
Glutamic acid	1.042 ± 0.055	1.130 ± 0.113	1.473	0.26
Cysteine	0.016 ± 0.003	0.026 ± 0.005	3.2	0.111
Arginine	0.534 ± 0.042	0.602 ± 0.065	0.3	0.599
Lysine	0.704 ± 0.068	0.772 ± 0.085	0.594	0.463
Tyrosine	0.378 ± 0.013	0.450 ± 0.041 *	5.763	0.043
Leucine	0.852 ± 0.054	0.938 ± 0.118	4.191	0.075
Proline	0.526 ± 0.035	0.514 ± 0.027	0.123	0.735
Serine	0.606 ± 0.030	0.700 ± 0.060	0.74	0.415
Threonine	0.492 ± 0.024	0.552 ± 0.054	1.635	0.237
Aspartic acid	1.922 ± 0.113	2.150 ± 0.237	1.833	0.213
Valine	0.626 ± 0.050	0.708 ± 0.121 *	5.906	0.041
Isoleucine	0.534 ± 0.048	0.576 ± 0.105 *	7.648	0.024
Histidine	0.264 ± 0.019	0.280 ± 0.031	1.428	0.266

## Data Availability

The data presented in this study are available in the article.

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
