# Peer review of "Effects of Sucrose Feeding on the Quality of Royal Jelly Produced by Honeybee Apis mellifera L."

_insects, 2023, doi:10.3390/insects14090742_

Round 1

Reviewer 1 Report

In their manuscript, the authors focused on the effects of artificial feeding, in particular sucrose feeding, on the production and quality of Royal Jelly in honey bees, Apis mellifera. Artificial feeding is an essential alternative for beekeepers during of shortage of nectar, which makes understanding honey or cane sugar feedings and their impact on colony development so important. The authors demonstrated experimentally that sucrose feeding has no significant effect on Royal Jelly production and/or quality. Overall, the authors provide practical data, especially for beekeepers, on the artificial diet of honey bees, which is generally lacking within the field.

However, I have some minor comments on the manuscript, which I have included below.

  • Line 11: Remove: “the industry”

  • Line 12: It is explicit that the experiment would use comparisons between sucrose-fed colonies and another group (control), but what is this control? It is a vague sentence.

  • Line 26: Again, it is vague information. What is the control group?

  • Line 55-56: Redundant sentence, as authors will cite each of these factors in the following sentences.

  • Line 66-69: Weird sentences. We know that nurse bees consume pollen (including beebread) to produce royal jelly for larvae, queens, and young adults. E.g., "Unlike nectar, pollen is unprocessed but is preserved by added honey bee secretions and low PH (Anderson et al., 2014). The nurse bee cohort consumes this pollen to produce royal jelly for the larvae, queen, and young adults (Crailsheim et al., 1992; Rortais et al., 2005). Royal jelly is also fed to foragers as a trigger to collect more or less pollen (Fewell & Winston, 1992; Camazine, 1993)." by Ivo Roessink & Jozef J. M. van der Steen (2021) Beebread consumption by honey bees is fast: results of a six-week field study, Journal of Apicultural Research, 60:5, 659-664.

  • Line 107-108: Remove: “At the end of the experiment,”

  • Line 132: I miss some information. What are these samples? What is the time point that authors collected these samples?

  • Line 157-159: Remove the entire paragraph.

  • Line 174-177: Redundant, confusing, rewrite.

  • Line 196-197: Authors can remove p-values.

  • Line 243-245: Remove the entire sentences. There is no connection between this sentence and the context (we know that feed impacts humans, but it is not the context)

  • Line 301: Remove “preliminary” (authors can say that this project is the 1st step to understanding the topic, but preliminary makes the manuscript weak)

Author Response

Dear Sir or Madam,

First, thank you for the recognition and praise of our work. We would like to thank you in particular for your professional and constructive comments on our work, which were used to improve our work and the quality of the manuscript. We have carefully discussed your suggestions and revised the relevant contents. We hope these revisions adequately address your concerns and can make our paper more favorable for publication. The revisions were addressed point-by-point below.

Yours sincerely,

Baohua Xu

Point 1: Line 11: Remove: “the industry”.

Response 1: The content has been deleted.

Point 2: Line 12: It is explicit that the experiment would use comparisons between sucrose-fed colonies and another group (control), but what is this control? It is a vague sentence.

Response 2: Thank you for pointing this out. The control group represents the honey-fed group, and we have made the necessary correction to this.

Point 3: Line 26: Again, it is vague information. What is the control group?

Response 3: We have revised this sentence.

Point 4: Line 55-56: Redundant sentence, as authors will cite each of these factors in the following sentences.

Response 4: Thank you for your comments. We have deleted this sentence.

Point 5: Line 66-69: Weird sentences. We know that nurse bees consume pollen (including beebread) to produce royal jelly for larvae, queens, and young adults. E.g., "Unlike nectar, pollen is unprocessed but is preserved by added honey bee secretions and low PH (Anderson et al., 2014). The nurse bee cohort consumes this pollen to produce royal jelly for the larvae, queen, and young adults (Crailsheim et al., 1992; Rortais et al., 2005). Royal jelly is also fed to foragers as a trigger to collect more or less pollen (Fewell & Winston, 1992; Camazine, 1993)." by Ivo Roessink & Jozef J. M. van der Steen (2021) Beebread consumption by honey bees is fast: results of a six-week field study, Journal of Apicultural Research, 60:5, 659-664.

Response 5: Thank you for your comments. The traditional belief is that nurse bees consume beebread before secreting royal jelly to feed the larvae. However, our preliminary research has found that during the peak flowering season, nurse bees may directly feed queen larvae with a small amount of beebread.

Reference: Wang, Y.; Ma, L.-T.; Xu, B.-H. Diversity in life history of queen and worker honey bees, Apis mellifera L. Journal of Asia-Pacific Entomology 2015, 18, 145-149, doi:https://doi.org/10.1016/j.aspen.2014.11.005.

Point 6: Line 107-108: Remove: “At the end of the experiment,”

Response 6: We have revised this sentence.

Point 7: Line 132: I miss some information. What are these samples? What is the time point that authors collected these samples?

Response 7: Thank you for your reminder. We have revised this sentence.

Point 8: Line 157-159: Remove the entire paragraph.

Response 8: Thank you for your reminder. We have deleted this paragraph.

Point 9: Line 174-177: Redundant, confusing, rewrite.

Response 9: Thank you for pointing this out. We have revised these sentences.

Point 10: Line 196-197: Authors can remove p-values.

Response 10: Thank you for pointing this out. All p-values have been removed.

Point 11: Line 243-245: Remove the entire sentences. There is no connection between this sentence and the context (we know that feed impacts humans, but it is not the context)

Response 11: Thank you for your comments. We have deleted this sentence.

Point 12: Line 301: Remove “preliminary” (authors can say that this project is the 1st step to understanding the topic, but preliminary makes the manuscript weak)

Response 12: Thank you for your comments. We have revised this sentence.

Reviewer 2 Report

The quality of royal jelly is primarily influenced by the abundance of pollen. This is either continuously brought into the hives or consumed by the bees from the stores in the combs. What is missing from the methodology is a comment on the level of pollen stocks in the experimental and control colonies and the pollen foraging during the experiment. 

Author Response

Dear Sir or Madam,

First, thank you for the recognition and praise of our work. We would like to thank you in particular for your professional and constructive comments on our work, which were used to improve our work and the quality of the manuscript. We have carefully discussed your suggestions and revised the relevant contents. We hope these revisions adequately address your concerns and can make our paper more favorable for publication. The revisions were addressed point-by-point below.

Yours sincerely,

Baohua Xu

Point 1: The quality of royal jelly is primarily influenced by the abundance of pollen. This is either continuously brought into the hives or consumed by the bees from the stores in the combs. What is missing from the methodology is a comment on the level of pollen stocks in the experimental and control colonies and the pollen foraging during the experiment.

Response: During this period, there were no external sources of nectar available, but there was a blooming season of plants that provided pollen sources. As a result, all bee colonies had ample pollen supply throughout the entire experimental period. We supplemented this information in the Materials and Methods section 2.1.

Reviewer 3 Report

I suggest that the authors do an experiment by feeding not with similar food, but by adding some innovative immunostimulating supplement to the sucrose. It can be herbal, containing vitamins or of their choice. Then see how it affects mothers' production and intake. There are many literary sources on the subject. For example Shumkova and colleagues 2021, 2022 if they wish they can include in the citations.

I think the english is good. 

Author Response

Dear Sir or Madam,

First, thank you for the recognition and praise of our work. We would like to thank you in particular for your professional and constructive comments on our work, which were used to improve our work and the quality of the manuscript. We have carefully discussed your suggestions. As you suggested and building upon prior research, we conducted an investigation in June 2023 to examine the impact of incorporating vitamins into sucrose on the production and caliber of royal jelly. Preliminary findings indicate that the inclusion of compliant vitamins in the sucrose substantially enhances both the quantity and quality of royal jelly, thereby yielding a noticeable rise in the vitamin content within royal jelly. We eagerly anticipate the expeditious completion and publication of this affiliated study. We are grateful for your valuable comment and suggestion.

Yours sincerely,

Baohua Xu

Round 2

Reviewer 1 Report

The authors did a good job of addressing all my concerns. I have no further comments.